# Trends in Prescription Opioid Use in Motor Vehicle Crash Injuries in the United States: 2014–2018

**DOI:** 10.3390/ijerph192114445

**Published:** 2022-11-04

**Authors:** Lan Jin, Sten H. Vermund, Yawei Zhang

**Affiliations:** 1Department of Neurosurgery, Yale School of Medicine, Yale University, New Haven, CT 06510, USA; 2Department of Epidemiology of Microbial Diseases, Yale School of Public Health, Yale University, New Haven, CT 06510, USA; 3National Cancer Center/National Clinical Research Center for Cancer/Cancer Hospital, Chinese Academy of Medical Sciences and Peking Union Medical College, Beijing 100021, China

**Keywords:** opioids, motor vehicle crash, trends, United States, non-fatal injury, emergency visits

## Abstract

Motor vehicle crashes (MVC) cause over three million people to be nonfatally injured each year in the United States alone. We investigated trends and patterns in prescription opioid usage among nonfatal MVC injuries in 50 states in the US and the District of Columbia from 2014 to 2018. All emergency department visits for an MVC event (*N* = 142,204) were identified from the IBM^®^ MarketScan^®^ Databases. Using log-binomial regression models, we investigated whether the prevalence of prescription opioids in MVC injuries varied temporally, spatially, or by enrollees’ characteristics. Adjusting for age, relationship to the primary beneficiary, employment status, geographic region, and residence in metropolitan statistical area, the prevalence decreased by 5% (95% CI: 2–8%) in 2015, 18% (95% CI: 15–20%) in 2016, 31% (95% CI: 28–33%) in 2017, and 49% (95% CI: 46–51%) in 2018, compared to 2014. Moreover, the prevalence decreased by 28% (95% CI: 26–29%) after the publication of the CDC Guidelines for Prescribing Opioids for Chronic Pain. Spatial variations were observed in the prevalence and temporal trend of prevalence. The decreasing trend in the prevalence of prescription opioids in MVC is consistent with the decrease in the dispensing rate of opioids and the percentage of high-dosage opioids in the study population.

## 1. Introduction

In the US, motor vehicle crashes (MVC) are a leading cause of death, especially for people aged less than 54 years [1]. Approximately three million people are nonfatally injured in MVC each year [2]. MVC-related injuries cause substantial economic burdens to society due to medical costs and productivity losses [3]. Exposure to opioids can cause drowsiness, confusion, nausea, impaired driving performance, and poor psychomotor outcomes (e.g., reaction time) [4,5,6]. A meta-analysis showed that people taking prescription opioids were more than twice as likely to be involved in MVC as those not taking them [7].

Among fatally injured drivers, the prevalence of prescription opioids increased sevenfold from the early 1990s to the early 2010s based on analyses of the US Fatality Analysis Reporting System (FARS) [8,9,10]. The most common opioids involved in MVC deaths, i.e., hydrocodone and oxycodone, showed a prevalence of increasing use in the early 2000s [8,9]. In the early 2010s, the Food and Drug Administration (FDA) and the Centers for Disease Control and Prevention (CDC) began to cite the serious risks of overdose or misuse associated with the long-term use of long-acting opioids, and the FDA took a series of measures toward controlling the appropriate prescribing practices of long-acting opioids [11]. In March 2016, the CDC published guidelines for Prescribing Opioids for Chronic Pain, including the selection of opioids, dosage considerations, and duration of treatment (“CDC Guidelines”) [12]. In 2017, the US Department of Health and Human Services declared the opioid crisis a public health emergency. The promulgation of these guidance documents and regulatory measures corresponded to a plateau in opioid use prevalence in MVC deaths after 2012 [8,10]. However, the temporal trends of opioid use in MVC deaths in more recent years were seldomly reported to assess the impact of opioid-related regulations and guidelines on driving safety. Additionally, as nonfatal injuries affect a broader population, it is important to understand the trends and patterns of opioid usage in MVC injuries, in addition to MVC deaths. However, the evidence of the trends in prescription opioid usage in nonfatal injuries is sparse.

Additionally, limitations exist in using FARS to detect the temporal trends of opioid use in MVC due to inconsistencies in drug testing and reporting procedures across states and jurisdictions [13]. Exploring additional sources of data is needed to better understand the temporal patterns of drug use in MVC. To address the gap in knowledge of the trends and patterns in the prevalence of prescription opioid use in nonfatal MVC injuries, we investigated changes in the prevalence of prescription opioids in MVC injuries from 2014 to 2018 and the spatial variations in the prevalence and temporal changes in the prevalence.

## 2. Materials and Methods

### 2.1. Study Population

We obtained health insurance claims data for enrollees residing in 50 states and the District of Columbia from the IBM^®^ MarketScan^®^ Databases from 2014 to 2018. The MarketScan Databases are one of the largest collections of de-identified patient-level data covering the continuum of care, including outpatient and pharmacy, from large employers and health plans across the US. The MarketScan Databases contain information on over 255 million enrollees since 1995, which is large enough to create a nationally representative sample of Americans with employer-provided health insurance [14].

In outpatient services databases, we identified all Emergency Department visits for an MVC event from 2014 to 2018 based on ICD-9 codes (E810–E825) and ICD-10 codes (V00–V89) [15,16]. We used an Agency for Healthcare Research and Quality toolkit, MapIT, to confirm the compatibility of the selected ICD-9 and ICD-10 codes [17].To be included in the analyses, the enrollees were ≥18 years and had at least 6 months of continuous enrollment by the time of their MVC injuries. Enrollees were excluded if their pharmaceutical claims were not captured by the MarketScan Databases.

From the pharmaceutical claims databases, we extracted the dispensing dates and days of supply for prescription opioids. Prescription opioids were identified based on the National Drug Code compiled by the CDC for analyzing population-level opioid prescriptions [18]. If an MVC injury occurred during the prescription period of any opioids, the MVC injury was deemed to be associated with prescription opioid use.

### 2.2. Statistical Analysis

We compared the characteristics of the study population with and without the involvement of prescription opioids. The enrollees’ characteristics included age, sex, insurance plan type, relationship to the primary beneficiary, and employment status of the primary beneficiary. For defining the time of MVC injuries, we investigated both (1) the year of the MVC injury (i.e., 2014–2018) and (2) a binary variable indicating whether an injury occurred before or after the publication of the 2016 CDC guidelines. Spatial variables included the geographic region of residence (Northeast, North-Central, South, or West) and whether or not an enrollee resided in a metropolitan statistical area (MSA). The differences in the enrollees’ characteristics and variations in temporal or spatial variables by the involvement of prescription opioids in MVC injuries were tested using Chi-square tests for the nominal variables and Cochran–Armitage trend tests for the ordinal variables.

Prevalence ratios (PR) were used to quantify whether the prevalence of prescription opioids in MVC injuries changed temporally (i.e., by the year of the accident or after versus before the CDC guidelines), spatially (i.e., by geographic region or by MSA versus non-MSA), or were associated with enrollees’ characteristics. The prevalence ratios were estimated using log-binomial regression models with a random effects variable for the state of residence of the injured party [9]. The random effects of state were included to account for the opioid-prescribing practices, which varied by state [19]. The enrollees’ characteristics that showed significant associations with prescription opioids in MVC injuries were included in the final models for the time variables (i.e., the year of the MVC injury and before-after the CDC guidelines). We also investigated the temporal trends in the use prevalence of common opioids (i.e., hydrocodone, oxycodone, tramadol, codeine, morphine, buprenorphine, fentanyl, and methadone) in MVC. As it may take time for the guidelines to change practice, we conducted sensitivity analyses to assess the impact of the guidelines after a transitory period. Specifically, we considered three scenarios of transitory periods (i.e., 3, 6, and 12 months) in estimating the prevalence ratios before versus after the CDC guidelines.

To investigate whether the temporal trends in prescription opioid use in MVC injuries varied by geographical region or between MSA and non-MSA, we tested the interactions between these spatial variables and the time variables in the log-binomial regression models. We summarized and mapped state-level prescription opioids in MVC injuries by year, and the state-level prevalence ratios in the before-after CDC guidelines comparison.

To explore the potential factors explaining the temporal trends in prescription opioid use in MVC injuries, we investigated the following dispensing practice measures: the dispensing rate of opioids (total number of prescriptions per 100 MVC injuries), total dosing volume of prescription opioids measured by morphine milligram equivalents (MME), daily MME, prescription duration, percentage of high-dosage opioid prescriptions (daily MME ≥ 90 mg) [19], and percentage of long-acting opioid prescriptions. These dispensing practice measures were compared between the states with high and low prevalence ratios (median as the cutoff) in the before-after CDC guidelines comparison.

Data preparation was conducted using SAS version 9.4 (SAS Inc., Cary, NC, USA), and statistical analyses were conducted using R version 4.0.3 (R Foundation for Statistical Computing, Vienna, Austria). All tests were 2-sided with a significance level of α = 0.05.

## 3. Results

We examined 142,204 emergency department visits for MVCs involving 137,839 enrollees from 2014 to 2018. Prescription opioids were involved in 19% of MVC injuries (*N* = 27,275). The characteristics of MVC injuries with and without the involvement of opioids were compared (Appendix A). In MVC injuries, prescription opioid usage was correlated with age, insurance type, relationship to the insured employee, and employment status (Student’s *t*-test or Chi-square test *p*-values < 0.001). Compared to those not involving opioids, MVC injuries involving prescription opioids showed a higher average age (41 vs. 38 years), similar proportions of fees for service insurance type (88% vs. 87%), a higher proportion of spouse of a primary beneficiary (23% vs. 16%), a lower proportion of children of the primary beneficiary (17% vs. 27%), and a lower proportion of active full-time employees (72% vs. 76%). There was no association between sex and opioid use in MVC injuries (*p*: 0.83).

The prevalence of prescription opioid usage in MVC injuries decreased continuously from 24% in 2014 to 12% in 2018 (Table 1). Accounting for the random effects of state, the prevalence decreased by 50% from 2014 to 2018 (PR: 0.50; 95% confidence interval [CI]: 0.48, 0.52). After the CDC guidelines, the prevalence of prescription opioid usage in MVC injuries decreased by nearly 30% (PR: 0.71; 95% CI: 0.70, 0.73). The results were consistent in sensitivity analyses considering transitory periods after the publication of the guidelines. The prevalence ratios comparing 3, 6, or 12 months after versus before the CDC guidelines were 0.69 (95% CI: 0.68, 0.71), 0.67 (95% CI: 0.65, 0.68), and 0.62 (0.60, 0.64), respectively (Appendix A).

Among common opioids, hydrocodone, oxycodone, tramadol, fentanyl, and methadone showed significant decreases after the CDC guidelines, whereas codeine, buprenorphine, and morphine did not show significant changes (Table 2). We illustrated the prevalence of common opioids in each year from 2014 to 2018 by plotting over time (Appendix A).

With additional adjustments for age group, relation to employee, employment status, geographic region, and MSA indicator, the decreasing trends in the prevalence of prescription opioid use in MVC injuries remained clear. Compared to 2014, the prevalence of use decreased by 5% in 2015 (PR: 0.95; 95% CI: 0.92, 0.98), 18% in 2016 (PR: 0.82; 95% CI: 0.80, 0.85), 31% in 2017 (PR: 0.69; 95% CI: 0.67, 0.72), and 49% in 2018 (PR: 0.51: 95% CI: 0.49, 0.54). The prevalence of use after the CDC guidelines decreased by 28% (PR: 0.72; 95% CI: 0.71, 0.74).

The prevalence of prescription opioid use in MVC injuries showed substantial variations at the regional level. The prevalence in the North-Central, South, and West regions was nearly twice as high as the prevalence in the Northeast region. Adjusting for the year of injury, age group, relation to employee, employment status, and MSA indicator, the regional variations in the prevalence were similar; compared to the Northeast region, the prevalence was significantly higher in the North-Central (PR: 1.74; 95% CI: 1.46, 2.07), South (PR: 1.79; 95% CI: 1.52, 2.10), and West regions (PR: 1.98; 95% CI: 1.66, 2.35). The prevalence was significantly higher in non-MSA than in MSA (PR: 1.04; 95% CI: 1.01, 1.08), but this association became non-significant in the final model. The decreasing trends in the prevalence varied by geographic region and urbanicity (i.e., MSA vs. non-MSA). Compared to MSA, non-MSA showed a smaller decrease after the CDC guidelines (*p* for interaction: <0.001). The prevalence of opioid use decreased by 29% (95% CI: 27–30%) in MSA and 21% (95% CI: 16–25%) in non-MSA. The decrease in the prevalence was smaller in the South (*p* for interaction: 0.005) and West (*p* for interaction: 0.012) than in the Northeast. We plotted the regional-level prevalence of prescription opioid use in MVC injuries for each year (Appendix A).

Between-state variations in the prevalence of prescription opioid use in MVC injuries during the 5 years increased nearly fourfold (range: 7.5% to 28.5%; median: 19.9%; Figure 1). We did not detect significant decreases in between-state variations over time. The median state-level prevalence decreased from 24% in 2014 to 11% in 2018 (Figure 1). We visualized the state-level prevalence in each year and prevalence ratios, comparing after and before the CDC guidelines on the map (Figure 2). The number of states in the highest quartile of prevalence (i.e., ≥24.6%) decreased from 25 states in 2014 to 1 state in 2018. The prevalence ratios after versus before the CDC guidelines showed a wide range (range: 0–1.07).

The decreasing temporal trends in the prevalence of prescription opioid use in MVC injuries were consistent with the decrease in dispensing rates (from 28 prescriptions per 100 injuries in 2014 to 15 prescriptions per 100 injuries in 2018) and the decrease in high-dosage prescriptions (from 6.2% in 2014 to 5.1% in 2018, among all opioid prescriptions) (Table 3). The total dosing volumes (i.e., MME), daily MME, duration, and long-lasting opioids did not decrease substantially over time. States with larger decreases (>29%) in the prevalence of opioid use in MVC injuries after the CDC guidelines showed larger decreases in dispensing rates (62% vs. 43% in other states) and larger decreases in high-dosage prescriptions (49% vs. 42% in other states) from 2014 to 2018.

## 4. Discussion

We found that the prevalence of prescription opioid use associated with MVC injuries decreased from 24% in 2014 to 12% in 2018 in 50 states and the District of Columbia. To the best of our knowledge, this is the first study reporting a decreasing trend in opioid-involved MVC injuries. A study on blood fentanyl concentrations in impaired drivers from 2014–2019 found that their prevalence increased from 4% to 22% in New Hampshire and from 7% to 40% in Florida; however, the illicit use of drugs was apparent in 60% of the fentanyl-involved cases [5]. In our study, we observed a 39% decrease in the prevalence of prescription fentanyl after the CDC guidelines in 2016, suggesting that policy measures may have been effective in reducing the involvement of prescription fentanyl in MVC injuries. The illicit use of opioids is hard to ascertain in fatally injured drivers and fentanyl concentrations in blood are hard to interpret [5]; therefore, the illicit use of opioids, including fentanyl, was seldomly reported in previous studies. As the illicit use of opioids is an important risk factor for impaired driving, future studies are needed to investigate the impact of the illicit use of opioids on driving safety. In Europe, toxicology testing has shown a wide range of opioids detected in suspected drugged drivers in different nations (i.e., 4% from 1998 to 2004 in Greece, 10% in 2005 in Switzerland, and 20% from 2003 to 2007 in Austria) [20,21,22]. There are methodological complexities in comparing the published results of toxicology testing since reports can vary by the type of biological sample and specific assay “yes-no” cutoffs [23]. A recent study on all road users who were seriously and/or fatally injured in MVC showed that the prevalence of opioid use increased after the COVID-19 public health emergency (before vs. after: 7.6% vs. 12.9%) [24]. Nonetheless, estimating the prevalence of opioid usage among people visiting Emergency Departments for an MVC event can provide an important perspective in understanding the exposures to opioids in nonfatal MVC injuries.

For fatal injuries, the prevalence of prescription opioid use increased about sevenfold from the early 1990s to the early 2010s and then plateaued up to 2016 [8,9,10]. Our analysis showed a 50% decrease from 2014 to 2018 for nonfatal injuries. Consistent with the previous study on fatal MVC injuries, [9] hydrocodone and oxycodone are the most commonly prescribed opioids, accounting for 70% of total opioids in nonfatal MVC injuries observed in our study. The prevalence of hydrocodone, fentanyl, and methadone use in MVC injuries showed larger decreases than other opioids.

Previous studies showed inconsistent results regarding the associations of demographic factors with the prevalence of prescription opioid use in MVC injuries. We found that the prevalence of prescription opioid use in MVC non-fatal injuries was higher in older age groups. Previous studies on fatal injuries did not find significant associations between age and opioid usage [8,9]. In a roadside voluntary survey of drivers, the prevalence of detected opioids was higher in the 45–64- than the 16–20-year age groups among nighttime drivers, whereas no difference in age groups was found among daytime drivers [25]. In our study, we did not detect differences in the prevalence of prescription opioid use in MVC injuries by sex, similar to findings in the roadside survey [25]. Other studies on fatal injuries have differed [7,9]. Based on FARS data, one study reported a significantly higher prevalence in male drivers in 24 states that conducted toxicological testing on more than 50% of fatally injured drivers, [9] whereas another study reported a significantly higher prevalence in female drivers in 6 states with a testing rate of over 80% [7]. Differences in population characteristics might explain the different results between the studies. Given these and other complexities, more work is needed to more fully understand other sociodemographic factors that might influence the link between opioid use and MVC injuries and fatalities.

We found that the prevalence of prescription opioid use in MVC injuries was lower in the Northeast than in the North-Central, South, or West regions. A previous study on Veterans Health Administration data in 2016 reported that opioid prescriptions measured by MME per capita were significantly lower in the Northeast than in the South regions [26]. At the state level, we also found a nearly fourfold variation in the prevalence of prescription opioid use in MVC injuries. This pattern was consistent with opioid-prescribing variations in the general population nationwide (two- to fivefold variation in various prescribing measures) [19,26,27]. Previous studies on fatal injuries have not investigated spatial variations in the prevalence of opioid usage. The roadside voluntary survey of drivers reported no significant regional variations in the prevalence of prescription opioid use [25]. Understanding the spatial patterns of prescription opioid use in MVC injuries can inform local and state policies on driving safety.

The temporal trends and spatial patterns of the prevalence of prescription opioid use in MVC injuries are reflected in the changing dispensing practices of opioids. From 2014 to 2018, we found a 46% decrease in the dispensing rate (number of opioid prescriptions per 100 MVC injuries), which was similar to the 50% decrease observed in the prevalence of opioid use in MVC injuries. This is consistent with the decrease in the dispensing rate for the general population nationwide from 75.6 per 100 people in 2014 to 51.4 in 2018 (32% decrease) [27]. In the population involved in MVC injuries, we found that the percentage of high-dosage opioid prescriptions decreased from 6.2% in 2014 to 5.1% in 2018, which was consistent with a decrease in high-dosage prescriptions in the general population nationwide (from 9.4% in 2014 to 8.5% in 2017) [19]. The states that showed larger decreases in the prevalence of prescription opioid use in MVC injuries after the CDC guidelines showed larger decreases in the dispensing rate and high-dosage prescriptions compared to the states with smaller decreases after the CDC guidelines were published.

We believe that the salutary trends noted in this study are hopeful signs that prescription opioid harm is diminishing along with declining prescriptions and dosages of MME. There are limitations to note in our study. Opioid-involved MVC injuries were defined as MVC injuries that occurred during an opioid prescription based on insurance claims databases so causality cannot be assured. Opioid involvement in MVC injuries determined by insurance claims data has been used to investigate MVC risks associated with opioids or other drugs [15,16]. Although we investigated a total of 142,204 MVC injuries in the US, the number of injuries per year is relatively small for some states, which prevented us from making inferences on the temporal trends in each state. However, we plotted the variation profiles and mapped the state-level prevalence by year so the variations in the prevalence across states and the overall temporal trends can be observed over the studied years. We cannot report on trends for non-insured persons in our databases, but the access to opioids, thus the prevalence of opioid use in MVC injuries, could differ by insurance status. We did not have information on the time of day of the accidents (e.g., nighttime vs. daytime), which modified the relationship between age and the prevalence of opioid use in a previous roadside survey on randomly selected drivers [25].

## 5. Conclusions

We found that the prevalence of prescription opioid use in MVC injuries decreased by 50% from 2014 to 2018, with a decrease of nearly 30% after the publication of the CDC Guidelines for Prescribing Opioids for Chronic Pain in March 2016 [12]. Spatial variations exist in the prevalence and the temporal trends of the prevalence. The changes in dispensing practices of opioids can partially explain the temporal trends and spatial patterns of prescription opioid use in MVC injuries. States concerned about continuing prescription opioid usage as a potential contributor to MVC injuries can improve surveillance of local dispensing practices, especially the number of prescriptions per capita and the percentage of high-dosage prescriptions. Our findings suggest that federal- and state-level efforts in curbing the opioid crisis have reduced opioid-involved MVC injuries.

## Figures and Tables

**Figure 1 ijerph-19-14445-f001:**
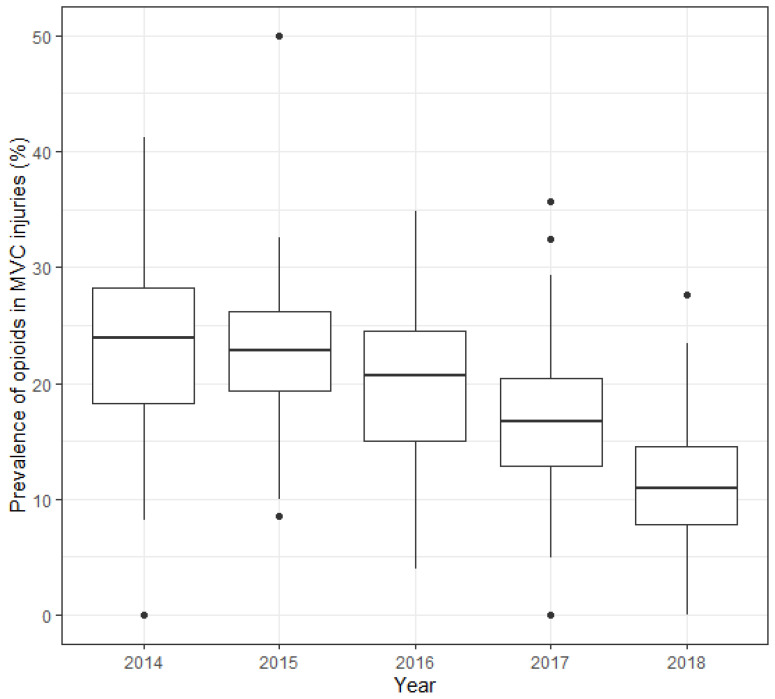
Between-state variations in prevalence of prescription opioid use in MVC injuries, US, 2014–2018. Abbreviation: MVC, motor vehicle crash.

**Figure 2 ijerph-19-14445-f002:**
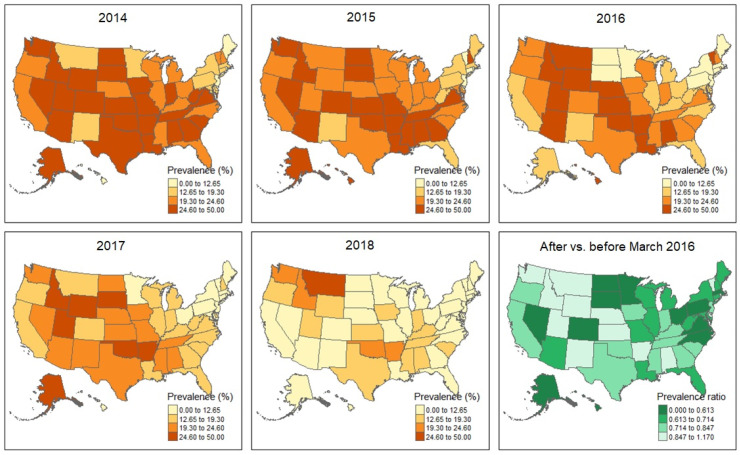
Prevalence of prescription opioid use associated with MVC injuries by state, 2014–2018. Abbreviation: MVC, motor vehicle crash. Darker brown color indicates a stronger MCV injury–opioid association. Prevalence ratios with the before-after March 2016 guidelines comparison (the release date for the Centers for Disease Control and Prevention Guidelines on safer and more effective opioids prescription in chronic pain treatment [12]); darker green color indicates a stronger putative influence of the CDC guidelines on reducing the involvement of opioids in MVC injuries.

**Table 1 ijerph-19-14445-t001:** Prevalence and prevalence ratios for prescription opioids in motor vehicle crash (MVC) injuries, US, 2014–2018.

	Number of MVC Injuries (*N* = 142,204)	Percentage of MVC Injuries Involving Opioids	Prevalence Ratio (95% CI) ^a^
Year of MVC injuries			
2014	20,818	24%	1 (Reference)
2015	33,452	23%	0.93 (0.90, 0.95)
2016	36,350	20%	0.80 (0.78, 0.83)
2017	27,673	16%	0.67 (0.65, 0.70)
2018	23,911	12%	0.50 (0.48, 0.52)
MVC injuries occurred before or after the CDC guidelines			
Before (<March 2016)	59,678	23%	1 (Reference)
After (≥March 2016)	82,526	16%	0.71 (0.70, 0.73)
Region			
Northeast	14,041	11%	1 (Reference)
North-Central	33,837	18%	1.70 (1.43, 2.02)
South	67,480	21%	1.83 (1.56, 2.14)
West	26,846	20%	2.01 (1.69, 2.39)
Metropolitan statistical area			
Yes	125,199	19%	1 (Reference)
No	17,005	21%	1.04 (1.01, 1.08)

Abbreviation: CDC, Centers for Disease Control and Prevention. ^a^ Prevalence ratios were estimated using log-binomial models that included univariate fixed effect and random effect of state.

**Table 2 ijerph-19-14445-t002:** Prevalence ratios for common prescription opioids after versus before CDC guidelines ^a^.

	Number of Prescriptions Involved in MVC Injuries, 2014–2018	Prevalence Ratio (after vs. before the CDC Guidelines in March 2016)
All opioids	31,584	0.71 (0.70, 0.73)
Common opioids		
hydrocodone	15,655	0.63 (0.61, 0.65)
oxycodone	6407	0.73 (0.69, 0.76)
tramadol	5635	0.79 (0.75, 0.83)
codeine	2445	0.98 (0.91, 1.07)
buprenorphine	438	1.15 (0.94, 1.42)
morphine	326	1.01 (0.79, 1.29)
fentanyl	217	0.61 (0.45, 0.82)
methadone	82	0.50 (0.31, 0.80)

Abbreviation: CDC, Centers for Disease Control and Prevention. ^a^ CDC guidelines on safer and more effective opioid prescription in chronic pain treatment were published in March 2016 [12].

**Table 3 ijerph-19-14445-t003:** Temporal and spatial variations in dispensing practices over time.

Year	Number of Prescriptions	Number of Prescriptions per 100 Injuries	MME	Daily MME	Duration (Days)	% High-Dosage ^a^ Prescription ^b^	% Long-Acting Prescription ^b^
All states							
2014	5811	28	828	73	9.2	6.2%	4.3%
2015	8389	25	638	83	8.9	5.9%	3.9%
2016	7903	22	857	107	9.4	5.4%	4.7%
2017	5812	21	499	57	9.4	5.3%	4.7%
2018	3669	15	893	66	9.6	5.1%	5.7%
States with larger decrease (PR ≤ 0.71 ^c^ after versus before the CDC guidelines)							
2014	2675	26	1273	108	9.3	7.2%	4.4%
2015	3977	24	728	116	8.8	5.1%	3.7%
2016	3438	20	1318	186	10.2	5.7%	5.4%
2017	2043	16	547	66	9.6	4.9%	5.4%
2018	1285	10	1730	105	10.8	3.7%	8.2%
States with smaller decrease (PR > 0.71 after versus before the CDC guidelines)							
2014	2906	28	455	44	9.2	5.5%	4.1%
2015	4372	26	557	53	8.9	6.8%	4.1%
2016	4446	23	504	46	8.8	5.2%	4.2%
2017	2872	20	396	48	8.7	3.9%	3.9%
2018	1778	16	395	37	8.8	3.1%	4.4%

Abbreviations: PR, prevalence ratio; MME, morphine milligram equivalent. ^a^ High-dosage prescriptions were defined as a daily dosage of 90 MME or more. ^b^ The percentages of high-dosage prescriptions and long-acting prescriptions were calculated among all prescriptions of opioids. ^c^ The median of the PR (after versus before the CDC guidelines in March 2016 [12]) among the 50 states and the District of Columbia was 0.71.

## Data Availability

Restrictions apply to the availability of these data. Data were obtained from IBW Watson Health and are available from the authors with the permission of IBW Watson Health.

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
