# Peer review of "Trends in Prescription Opioid Use in Motor Vehicle Crash Injuries in the United States: 2014–2018"

_ijerph, 2022, doi:10.3390/ijerph192114445_

Round 1

Reviewer 1 Report

This work is very interesting but the conclusions reached are quite weak and inconclusive: “Our findings suggest these trends to be useful surrogates for trends in the opioid. prevalence in MVC injuries. "

The presentation of the data is detailed. Some tables could be improved, such as table 1, which is very long and not easy to read, while figure 1 is not easy to understand.

Lines 211-212: the point discusses the use of illicit opiates and is very important and must be discussed more fully

Lines 233-269: the paragraph is a list of data collected from different studies that lead to inconsistent conclusions. A comment from the authors at the end of this presentation would be helpful.

I think it useful to underline one of the limitations of the study reported in lines 282-283.

The bibliography is complete

Reviewer 2 Report

This is a well written paper.  However, I don’t think it adds a lot of new knowledge to the field.  The authors should focus on what is new in their data.

The authors dichotomize the time periods to before and after the issuance of the CDC guidelines.  Given that it would take a period of time for the guidelines to change practice, it would be useful to include transitory period for a number of months after the issuance of the guidelines in the analysis. 

It would be clearer to present row per centages instead of column percentages in table e1.
